# Single-cell transcriptomic analysis in a mouse model deciphers cell transition states in the multistep development of esophageal cancer

Jiacheng Yao [1,8], Qionghua Cui [2,8], Wenyi Fan[2,8], Yuling Ma [2,8], Yamei Chen [2,8], Tianyuan Liu [2,8], Xiannian Zhang[3], Yiyi Xi[2], Chengcheng Wang[2], Linna Peng[2], Yingying Luo [2], Ai Lin[2], Wenjia Guo[2], Lin Lin[2], Yuan Lin[4], Wen Tan[2], Dongxin Lin[2,6,7✉], Chen Wu [2,5,6✉] & Jianbin Wang [1✉]

Esophageal squamous cell carcinoma (ESCC) is prevalent in some geographical regions of the world. ESCC development presents a multistep pathogenic process from inflammation to invasive cancer; however, what is critical in these processes and how they evolve is largely unknown, obstructing early diagnosis and effective treatment. Here, we create a mouse model mimicking human ESCC development and construct a single-cell ESCC developmental atlas. We identify a set of key transitional signatures associated with oncogenic evolution of epithelial cells and depict the landmark dynamic tumorigenic trajectories. An early down-regulation of CD8+ response against the initial tissue damage accompanied by the transition of immune response from type 1 to type 3 results in accumulation and activation of macrophages and neutrophils, which may create a chronic inflammatory environment that promotes carcinogen-transformed epithelial cell survival and proliferation. These findings shed light on how ESCC is initiated and developed.

[1] School of Life Sciences and Tsinghua-Peking Center for Life Sciences, Tsinghua University, Beijing, China. [2] Department of Etiology and Carcinogenesis, National Cancer Center/Cancer Hospital, Chinese Academy of Medical Sciences (CAMS) and Peking Union Medical College (PUMC), Beijing, China. [3] School of Basic Medical Sciences, Beijing Advanced Innovation Center for Human Brain Protection, Capital Medical University, Beijing, China. [4] Beijing Advanced Innovation Center for Genomics (ICG), Biomedical Pioneering Innovation Center (BIOPIC), School of Life Sciences, College of Engineering, and Peking-Tsinghua Center for Life Sciences, Peking University, Beijing, China. [5] Collaborative Innovation Center for Cancer Personalized Medicine, Nanjing Medical University, Nanjing, China. [6] CAMS Oxford Institute (COI), Chinese Academy of Medical Sciences, Beijing, China. [7] Sun Yat-sen University Cancer Center, State Key Laboratory of Oncology in South China, Guangzhou, China. [8] These authors contributed equally: Jiacheng Yao, Qionghua Cui, Wenyi Fan, Yuling Ma, Yamei Chen, Tianyuan Liu. ✉email: lindx@cicams.ac.cn; chenwu@cicams.ac.cn; jianbinwang@tsinghua.edu.cn

Esophageal squamous cell carcinoma (ESCC) is a common gastrointestinal malignancy in some parts of the world like China, with a poor prognosis and high mortality mainly due to the lack of specific measures for early diagnosis and effective therapies. ESCC represents an extraordinary paradigm of carcinoma development, shaped in a sequential manner from inflammation (INF), hyperplasia (HYP), dysplasia (DYS), carcinoma in situ (CIS) to invasive carcinoma (ICA)[1]. However, up to date, how ESCC initiates and develops is largely unknown. This long-standing question is the major factor obstructing the early intervention and clinical care improvement of the disease. As such, exploring the mechanisms underlying ESCC formation and identifying biomarkers are the crucial tasks for early detection, diagnosis, and precision treatment of the cancer.

Recent genome studies, including The Cancer Genome Atlas (TCGA) project, have identified many genome variations in ESCC by using whole-exome or whole-genome sequencing on clinical tissue samples[2,3]. Although these studies have revealed an important role of the identified genome alterations in ESCC, it remains unresolved that how the normal epithelial cells may transit by the mutations through precancerous lesions to invasive carcinoma because all these previous studies were in cross-sectional design. Another important issue in this regard is that somatic mutations solely might not be sufficient for ESCC initiation and development because such mutations also occur in pathologically normal human esophageal tissues[4]. These findings imply that other mechanisms such as transcriptome aberrance in ESCC tumorigenesis may merit further investigation. In addition, it has been demonstrated that the complex context of tumor microenvironment (TME) also has important roles in tumor initiation and development. Therefore, elucidating dynamic transcriptomic changes of TME cellular compositions during tumorigenesis are significant and inevitable in discovering how ESCC develops. Recently established single-cell transcriptomic analysis is a promising approach, which allows to analyze the complex cellular compositions and to decipher cell state transition in tissue samples[5].

Capturing continuous tumorigenic lesions from sole patient over time to perform such study is impossible and, therefore, to address this important issue would be highly dependent on well-established animal models. Fortunately, it has been shown that chemical carcinogen 4-nitroquinoline 1-oxide (4NQO) can induce mouse ESCC development in a manner that mimics the tumorigenic processes of ESCC in humans[6,7]. The distinct multiple stages of ESCC tumorigenesis induced by the carcinogen provide an excellent opportunity to interrogate cell state transition by single-cell RNA sequencing (scRNA-seq), which would elucidate the dynamic ESCC tumorigenesis.

Here we report a single cell-based transcriptomic profiling study on various types of cells across every pathogenic stage during ESCC initiation and development in a mouse model induced by 4NQO. We have built a complete atlas and characterized the transcriptomic profiles for the transition of esophageal epithelial cells under the attack of carcinogen and elucidated how they evolve over time in a holistic approach. We have also depicted the transition landscapes of non-epithelial cells, i.e., fibroblasts and immune cells, in the esophageal microenvironments of different stages of tumorigenesis. Furthermore, we have found that some key changes in mice also occur in human esophageal tissue samples. These results shed light on the phenotypes and transition fates of different cell types across the tumorigenic processes of ESCC in animal model and may be implicated in human ESCC.

## Results
**Bulk RNA-seq and scRNA-seq of mouse esophageal samples.** To explore the transcriptomic alterations at various pathological stages during ESCC tumorigenesis, 8-week-old female C57BL/6 mice were treated with 4NQO for 16 weeks, which resulted in five recognizable precancerous and cancerous lesions in the esophagus, i.e., INF, HYP, DYS, CIS, and ICA (Fig. 1a; Supplementary Fig. 1a). We examined mice receiving 4NQO and found they all developed the expected lesions in the esophagus at different time points of experiment. The ESCC number per animal (mean ± SD) at week 26 was 6.0 ± 3.6. We first performed conventional RNA-seq on mini-bulks of normal epithelial samples obtained by laser-capture microdissection (LCM) from control mice at different ages of 1, 2, 8, and 25 months and various precancerous or cancerous lesions from 4NQO-treated mice. Principal component analysis (PCA) of the differentially expressed genes showed that the expression programs in mice exposed to 4NQO were substantially different from that in control mice; however, there existed some overlaps in the expression profiles between 4NQO-exposed and non-exposed mice. For instance, the transcriptomic profile for stage INF in 4NQO-exposed mice was similar to that in control mice aged 25 months (Fig. 1b), indicating that conventional RNA-seq could not precisely clarify the path of malignant cell transition during the development and progression of ESCC due to high intra-tissue heterogeneity.

We therefore conducted a time-ordered single-cell transcriptomic profiling on various esophageal lesions in 4NQO-exposed mice. For stage NOR or stage INF, we used the whole esophagus from 30 or 20 mice. For other stages, we used the lesion foci and the sample numbers were 32 HYP from 25 mice, 24 DYS from 17 mice, 24 CIS from 20 mice and 25 ICA from 23 mice, respectively (Fig. 1c). A total of 66,089 cells including 29,975 CD45[+] immune cells and 36,114 CD45[−] non-immune cells were obtained across various pathological stages (Fig. 1c; Supplementary Fig. 1b). The median unique molecular identifier (UMI) per cell was 7748 in immune cells and 9370 in non-immune cells (Supplementary Fig. 1c); with a median signal detect ability of 1936 genes for immune cells and 2620 genes for non-immune cells (Supplementary Fig. 1c), respectively. Based on the expressions of canonical markers, we classified immune cells into T cells, B cells, myeloid cells and natural killer cells (Fig. 1c; Supplementary Fig. 1d, e) and identified four clusters of non-immune cells including epithelial cells, fibroblasts, endothelial cells and myocytes by using t-distributed stochastic neighbor embedding (tSNE) (Fig. 1c; Supplementary Fig. 1f, g).

**Identifying epithelial cell types during ESCC tumorigenesis.** To discover how normal esophageal epithelium develops into invasive carcinoma, we next examined the expression alterations and functional changes in epithelial cells during the transition from normal to precancer or cancer. We identified 1756 epithelial cells across all six stages that were classified into six subtypes designated as EpiC 1 to EpiC 6 (Fig. 2a; Supplementary Fig. 2a and Supplementary Table 1). Through the analysis of pathway activities (Fig. 2b; Supplementary Fig. 2b), we found that EpiC 1 ($n = 339$) had higher expression of genes (e.g., Birc5, Mki67, Top2a, and Ube2c) in mitosis and proliferation[8–11] compared with other cell clusters, probably reflecting the ability of routine self-renewal of the esophageal epithelium. EpiC 2 ($n = 463$) had significantly higher expression of genes such as Gstm1, Gstm2, Adh7, and Aldh3a1, which were involved in the xenobiotic detoxification, likely due to the 4NQO exposure[12–14]. EpiC 3 ($n = 421$) had substantially higher expression of genes in response to extracellular carcinogen stimuli, pro-inflammation and Hif-1 signaling (e.g., Jun, Junb, Zfp36, and Atf3)[15–18]. EpiC 4 ($n = 110$) had a higher expression program for keratinization (e.g., Krt4, Krt13, S100a8, and S100a9)[19,20]. EpiC 5 ($n = 213$) featured by the upregulation of genes in the epithelial–mesenchymal transition (EMT) pathways (e.g., Mmp13

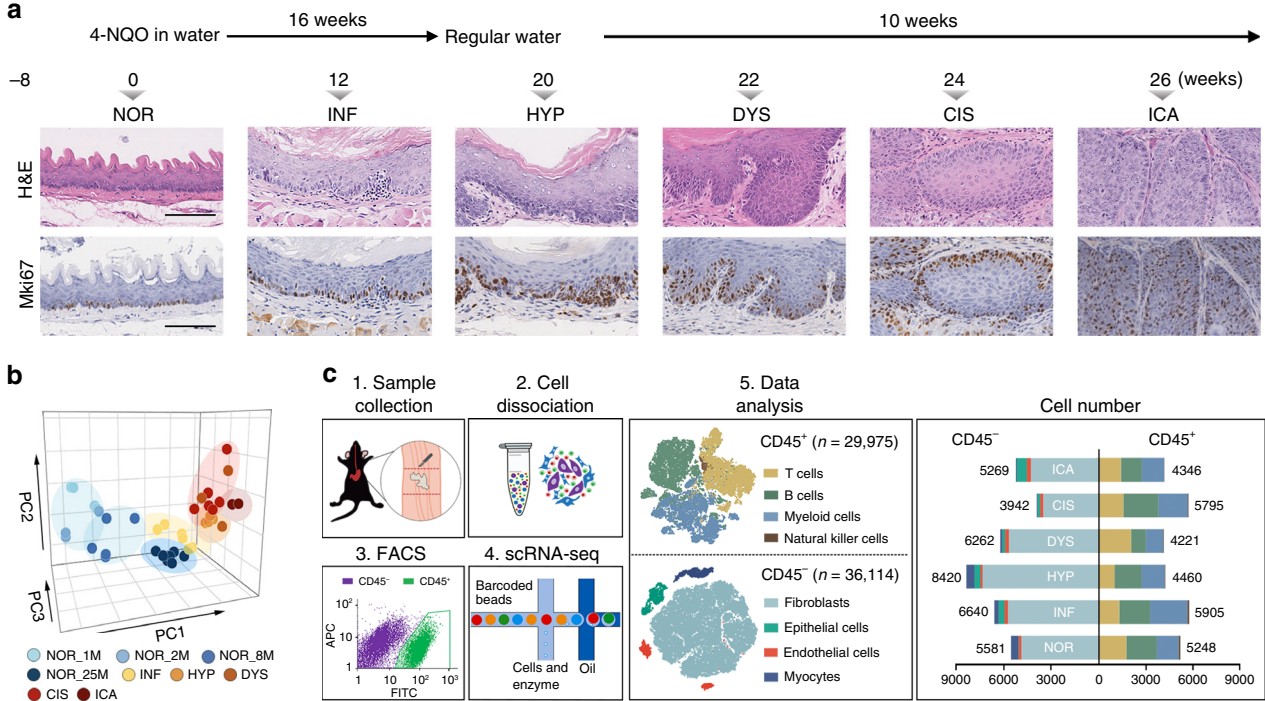

**Fig. 1 Experimental design of RNA-seq on 4NQO-induced esophageal lesions in mice. a** Induction of esophageal precancerous and cancerous lesions in mice. Mice were treated with 4NQO in drinking water (100 μg/ml) for 16 weeks and then kept without 4NQO treatment for another 10 weeks (upper panel). Mice were killed before (week 0), during (week 12) and after treatment (weeks 20, 22, 24, or 26), respectively. Hematoxylin–eosin (H&E) staining and immunohistochemistry (IHC) analysis of Mki67 on esophageal epithelium slides clearly identified six different pathological lesions, i.e., normal (NOR), inflammation (INF), hyperplasia (HYP), dysplasia (DYS), carcinoma in situ (CIS), and invasive carcinoma (ICA) (lower panel). Similar staining results were observed in over three visual fields from each stage of esophageal lesions (more staining image in Supplementary Fig. 2d). Scale bar, 100 μm. **b** Plot of principal component analysis (PCA) of mini-bulk tissue RNA-seq on different pathological lesions indicated by different colors. M month of mouse age. **c** Overview of the experimental design of scRNA-seq. Pathological lesions of the esophagus were dissected and digested into single-cell suspensions for further separation using FITC-CD45 antibody via FACS (1–4). CD45+ and CD45− cells whose numbers in different lesions are shown on right panel were scRNA-sequenced, respectively.

and *Itga6*[21,22] and angiogenesis (e.g., *Vegfa*)[23]. EpiC 6 ($n = 210$) had significantly upregulated expression in the genes controlling cancer invasiveness (e.g., *Mmp14* and *Ecm1*)[24,25] and metastasis (e.g., *Tm4sf1* and *Vim*)[26,27]. Besides, EpiC 5 cells also showed an expression pattern resembling the combination of EpiC 3, EpiC 4, and EpiC 6.

Three subtypes, EpiC 1–3, were all present in normal esophageal epithelium and at the various stages of 4NQO-induced abnormal epithelia, although the proportions of cells at each stage were different (Fig. 2c), suggesting that these subtypes of epithelial cells were the background components of esophageal tissues. EpiC 4 and EpiC 5 appeared from INF and HYP, respectively, while EpiC 6 was present uniquely at ICA as the major cell subtype (Fig. 2c). These cell subtype dynamics well concorded with the transitional role of EpiC 5 and the malignancy of EpiC 6 and prompted us to examine the cell status transition more closely. A single-cell diffusion map based on the gene expression similarity revealed a clear cell status transition across all pathological stages (Fig. 2d; Supplementary Fig. 2c). Gene expression vectors pointed from stage NOR cells to the centroid of cells from each stage. It was clear that major directional changes happened before the HYP and ICA stages, whereas the transition from stage HYP to stage CIS was mainly the expansion along the same direction. Together, these results indicated an important role of INF to HYP transition in ESCC development.

Further analysis revealed that the six representative genes selected from each epithelial cluster were differentially distributed among disease stages (Fig. 2e). *Aldh3a1* and *Atf3* were expressed across all stages and the levels were significantly higher at stage INF than that at the advancing stages. High level of *S100a8* appeared at stage HYP and covered all precancerous and ICA stages whereas the highest levels of *Itga6* and *Mmp14* were seen at stage ICA although their expressions were also detected in cells across all stages. The dynamic expressions of these genes at protein level in mice esophageal tissues with different disease stages were compared by using immunohistochemistry and the results were generally in line with their RNA expression despite of some disparity (Fig. 2f; Supplementary Fig. 2d). The abrupt upregulation of S100a8 in cells at stage HYP suggests a dramatic transition related to immune response.

**Identifying cell fates of epithelial cell status transitions**. We performed pseudotime and PCA analysis and found two evolution fates of esophageal epithelial cells during ESCC tumorigenesis both starting from EpiC 1 cells that had the lowest pseudotime value. Some cells transformed from proliferative EpiC 1 to normal differentiated EpiC 4 while other cells transformed to malignant EpiC 6, processing through EpiC 2 to EpiC 5 cells (Fig. 3a; Supplementary Fig. 3a, b). The evolution of EpiC 1 to EpiC 6 was mainly along component 1. Gene set variation analysis (GSVA) of component 1 revealed a significant enrichment of genes related to cell invasiveness, EMT and angiogenesis (Fig. 3b, c; Supplementary Fig. 3c), which was concordant with the expression programs of EpiC 6 cells (Fig. 2b). As EpiC 6 cells appeared only at the ICA stage, these results implied that

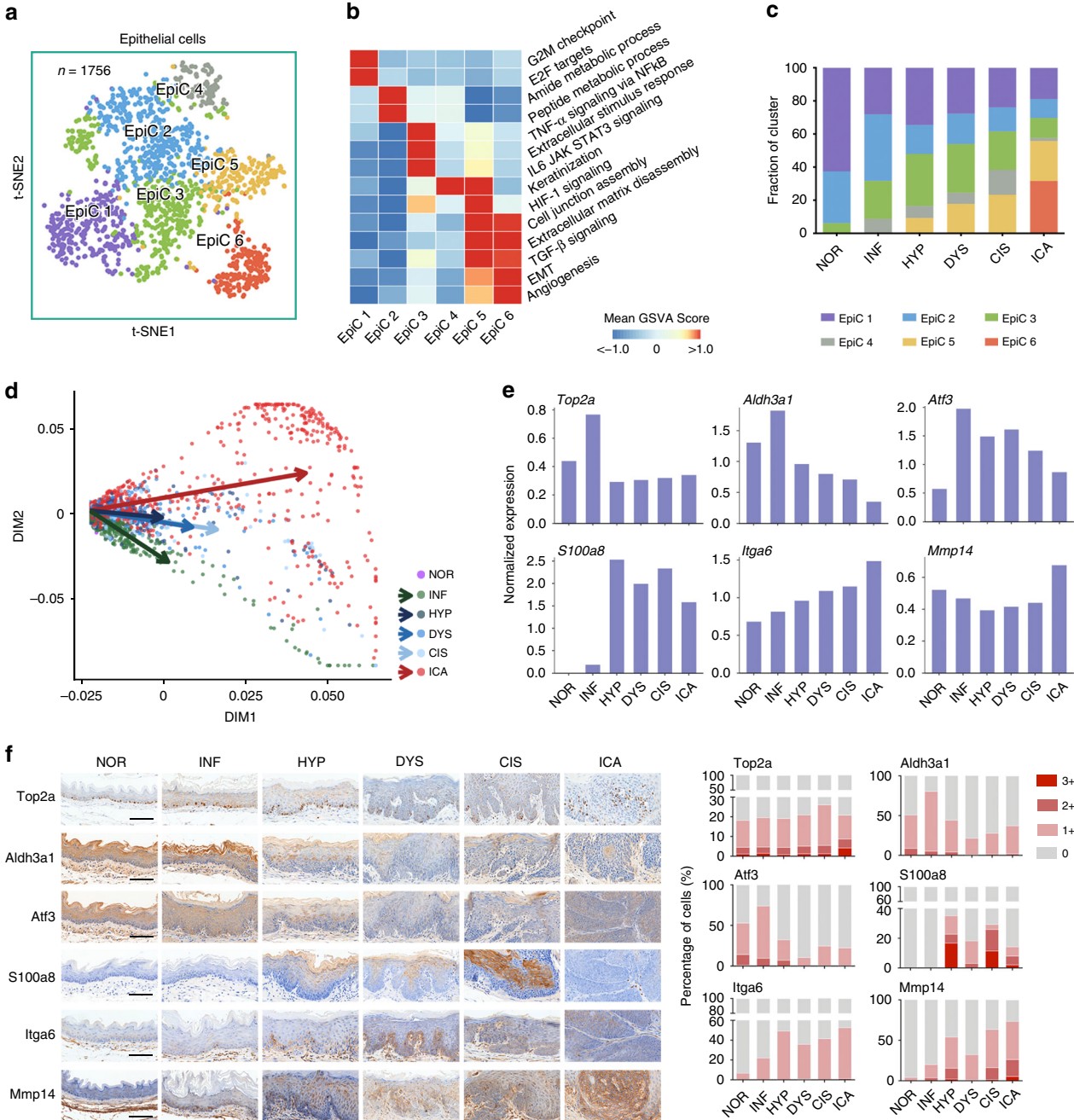

**Fig. 2 Distinct epithelial cell populations and their expression signatures. a** tSNE plot of 1756 epithelial cells based on their different expression. **b** Heatmap showing pathway activities scored per cluster by GSVA. **c** Stacked histogram showing epithelial cell composition across the six pathological stages. **d** Diffusion map of all epithelial cells using the first two diffusion components. Dot represents single cell and arrows indicate five branches starting from stage NOR to the other stages. **e** Histograms of scale normalized expression levels of six representative genes in each pathological stage. **f** Left: IHC staining of protein levels produced by the six genes in mouse esophageal tissues with different lesion. Scale bars, 100 μm. Right: stacked histograms showing quantification of IHC staining scores (0, negative; 1+, weak positive; 2+, median positive; 3+, strong positive). Each column was summarized from at least three visual fields.

component 1 might be the underlying molecular mechanism for malignant transition of the esophageal tissues (Supplementary Fig. 3d).

We then examined whether the alterations of any transcription factors (TFs), well-documented ESCC-related mutation, or methylation dysregulation were included in the oncogenic evolution along component 1. We found that the expression levels of *Pitx1*, *Trp53*, and *Bclaf1*, which are known tumor suppressor TFs[28–30], were substantially decreased while TFs

promoting EMT (e.g., *Ets1* and *Snai3*)[31,32], immunosuppression (e.g., *Eomes*)[33], cell migration and invasion (e.g., *Creb3* and *Elk3*)[34,35] were significantly upregulated. We also found two previously reported methylation-regulated genes in human ESCC, *Rab25*[36], and *Met*[37], were substantially down- or upregulated, respectively, along component 1. Furthermore, the expression levels of *Notch1*, a tumor suppressor frequently mutated in human ESCC[38], displayed a gradual decrease over time along the evolution of component 1 (Fig. 3d; Supplementary Fig. 3e).

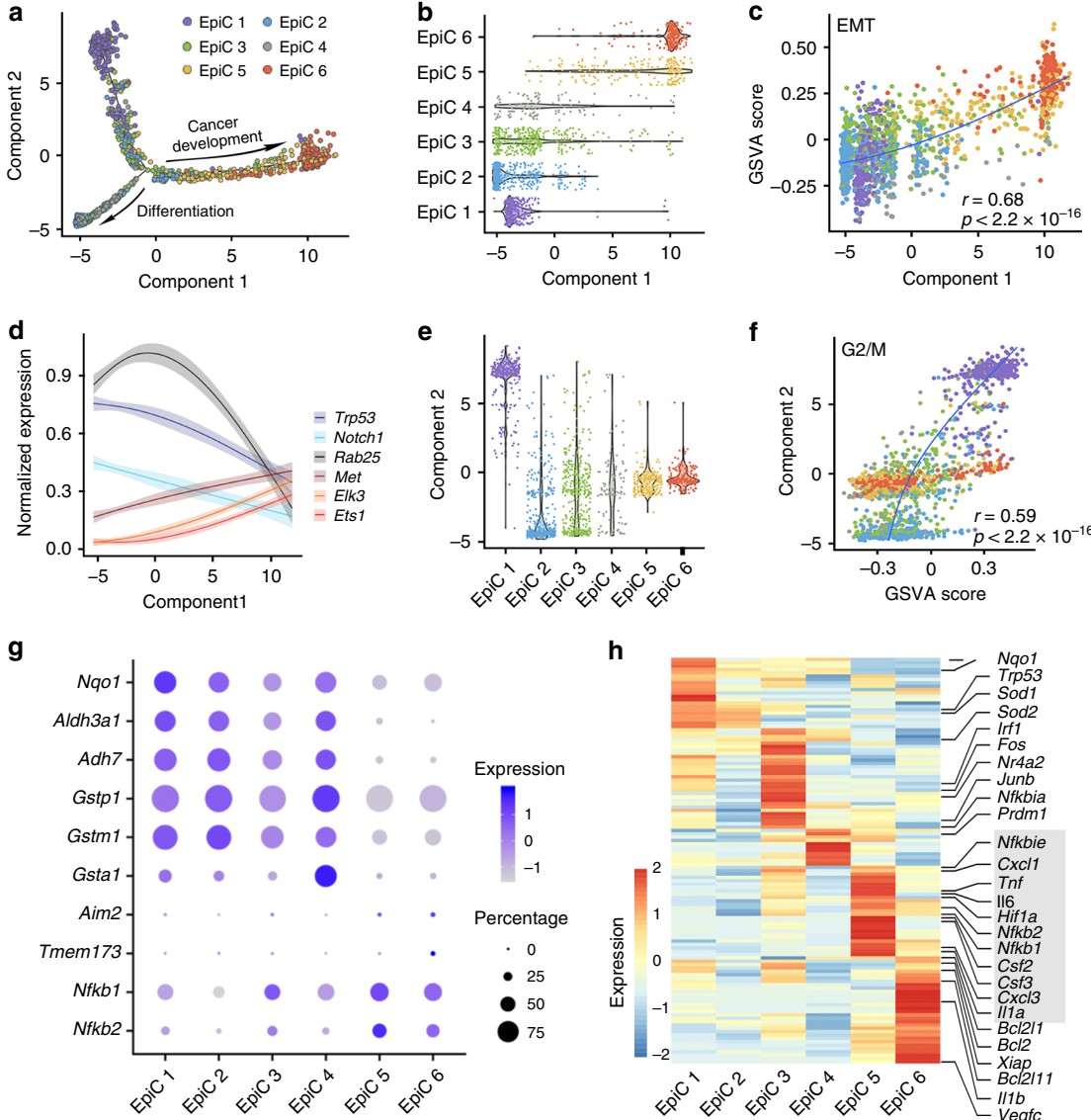

**Fig. 3 Characterization of epithelial cell transitions and key pathway changes. a** Pseudotime trajectory over epithelial cells in a two-dimensional statespace. Cell orders are inferred from the expression of the most dispersed genes across epithelial cell populations. **b** Violin plots of the distribution of the component 1 values across epithelial clusters. **c** Correlation between EMT pathway enrichment scores and component 1 values of single cells. **d** Normalized expression of six selected ESCC driver genes, methylation dysregulation genes, and transcription factors, smoothed over pseudotime component 1 using LOESS regression. Shaded regions indicate 95% confidence interval with a line indicating the mean gene expression. **e** Violin plots of the distribution of the component 2 values among sub-clusters. **f** Correlation between G2/M pathway enrichment scores and component 2 values of single cells. **g** Bubble plot showing expression levels of the genes related to response to 4NQO treatment across six cluster. Size of dots represents the percentage of cells expressing the gene; color scale shows the average expression level. **h** Heatmap displaying scale normalized expression level of genes in NF-κB signaling across the six epithelial clusters.

Genes in component 2 were mainly related to cell proliferation (e.g., *Birc5*, *Mki67*, and *Top2a*). Cell cycle regression analysis and GSVA scoring revealed a correlation between proliferative ability and value of component 2 (Fig. 3e, f). We found that EpiC 6 cells had the highest expression of proliferation-related genes and the highest proportion of cells at G2/M status as compared with other clusters except EpiC 1 cells (Fig. 3e, f; Supplementary Fig. 3f–h), indicating increased proliferative ability of this cell cluster. On the other hand, EpiC 4 cells were at the G1 and S status (Supplementary Fig. 3g, h), showing reduced proliferative capacity. Together, these unsupervised analyses depicted two clear cell fates during ESCC carcinogenesis and indicated that attacked by the carcinogen, a portion of esophageal epithelial cells went into oncogenic route (from EpiC 1 and EpiC 2 to EpiC 6)

but not into normal differentiation route (from EpiC 1 and EpiC 2 to EpiC 4). To exclude the impact of potential uncertainty of pathologically staging of early lesions such as hyperplasia and dysplasia, we also performed a sensitive analysis by excluding cells in week 20 or week 22. As a result, the cell state transition was similar to that without the exclusion (Supplementary Fig. 3i, j).

We further investigated the key pathways driving epithelial cells from stage INF to stage HYP and found that non-malignant EpiC 1–4 epithelial cells had significantly elevated expressions of the genes involving in carcinogen detoxification such as *Nqo1*, *Aldh3a1*, and *Gstp1* (Fig. 3g), which reflected normal cellular response to the damage induced by 4NQO. The continuous damage might induce immune response via the stimulator of

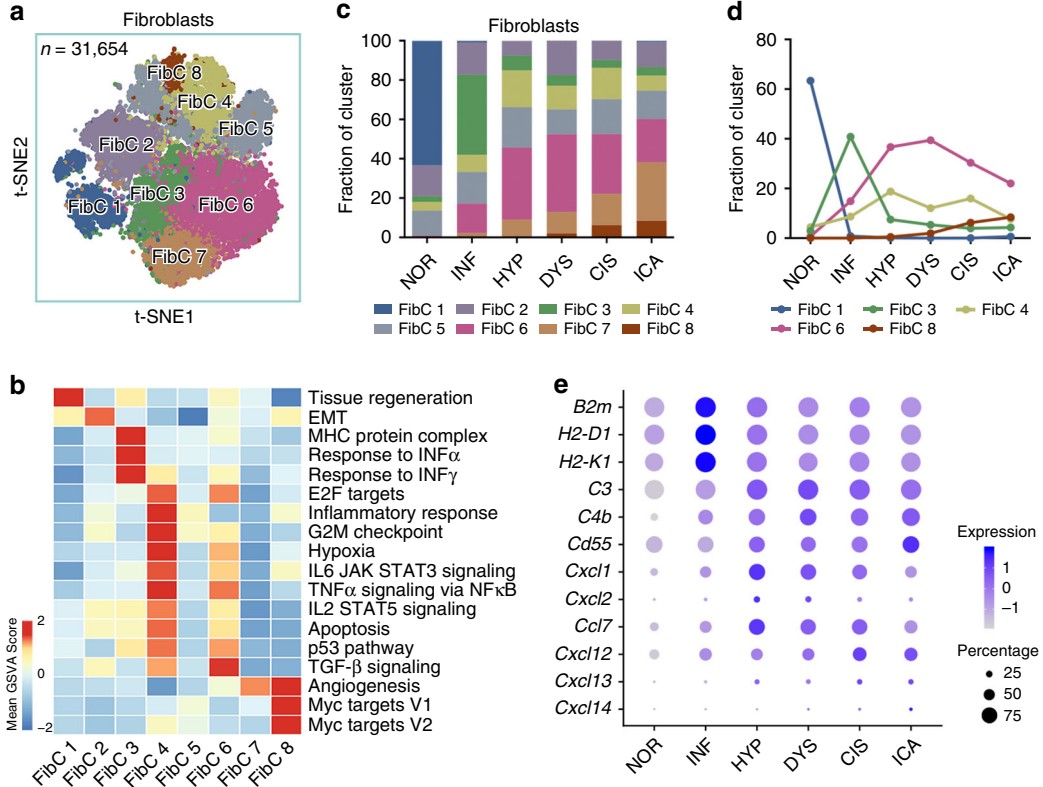

**Fig. 4 Identification of fibroblast clusters and their expression features. a** tSNE plot of 31,654 fibroblasts, colored by cluster. **b** Expression-based pathway activities scored by GSVA per fibroblast cluster. **c** Stacked histogram showing fibroblasts composition across the 6 pathological stages. **d** Line chart displaying changing trend of proportion of the four selected clusters across the six pathological stages. **e** Bubble plot showing scale normalized expression of representative genes involved in cytokine secretion, complement activity, and antigen presentation. Size of dots represents percentage of cells expressing corresponding genes in the cluster.

interferon genes (STING), AIM2 and NF-κB signaling because the expression levels of *Tmem173*, *Aim2*, *Nfkb1*, and *Nfkb2* were significantly elevated (Fig. 3g). In addition, we observed substantial differences in the expressions of NF-κB downstream genes in epithelial cell clusters (Fig. 3h). EpiC 3 cells had an increased expression of some transcription-related genes such as *Junb* and *Fos*, whereas EpiC 5 cells showed higher expression levels of inflammation genes (e.g., *Cxcl1*, *Il6*, *Tnf*, and *Csf3*) compared with other EpiCs, implying that they had crosstalks with immune cells. EpiC 5 cells also had high level of *Hif1a*, which may lead to metabolism remodeling that promoted cell transformation. As malignant cells, EpiC 6 showed strong activation of survival-related genes (e.g., *Bcl2* and *Xiap*) (Fig. 3g, h). These results indicated that under 4NQO attack, some epithelial cells might endure the damage and activate the inflammatory response, resulting in consequent cell survival and transformation.

**Transcriptomic changes in fibroblasts promote tumorigenesis.** We next examined the transcriptomic alterations of cells in the tissue microenvironment and identified eight fibroblast clusters (FibCs) by different gene expression patterns (Fig. 4a; Supplementary Fig. 4a, b) that showed similar expression pathways and dynamics to epithelial cells (Fig. 4b–d). FibC 1 was the dominant composition in stage NOR and then markedly decreased with the processing of tumorigenesis (Fig. 4c, d). FibC 3 cells were the major fibroblasts in stage INF and had a strong interferon response and MHC-mediated antigen presentation activity (Fig. 4b), representing the initial immune response to tissue damage. FibC 4 and FibC 6, the major fibroblasts in stage DYS,

had the elevated expression of genes involved in the TGF-β, STAT5 induced by IL-2 or STAT3 induced by IL-6 signaling pathways. FibC 8 cells increased along the tumorigenic stages with high expressions of genes in *Myc* and angiogenesis pathways. The replacement of FibC 3 by FibC 4 and FibC 6 during INF to HYP transition further confirmed the shift of immune response during early ESCC development. Specifically, beginning from stage HYP, fibroblasts actively recruited immune cells through increasing the expressions of complements and chemokines (Fig. 4e). For example, we found that FibC 6 and FibC 8 had significantly elevated expressions of various chemokine-related genes (e.g., *Ccl7* in FibC6 and *Cxcl12* in FibC 8, Supplementary Fig. 4c), which were well-known molecules that recruit immune cells. The dynamic changes of fibroblast clusters suggested that the immune response is modulated during ESCC tumorigenesis.

**Turndown of adaptive anticancer immune during tumorigenesis.** We then explored the T cell status during ESCC tumorigenesis and identified 4 clusters designated as CD8[+] T cell, CD4[+] T cell, CD4[−]CD8[−] T cell 1 and CD4[−]CD8[−] T cell 2, respectively (Fig. 5a). We further classified CD8[+] T cells into 7 clusters (Fig. 5b) and CD4[+] T cells into 7 clusters (Fig. 5c), respectively. During 4NQO-induced tumorigenesis, all CD8[+] T cells expressed naive and memory marker genes (e.g., *Tcf7*, *Il7r*, and *Ccr7*; Supplementary Fig. 5a)[39–41] rather than cytotoxic markers. In stage NOR, the major type was naive cells (CD8-TN1) while in stage INF, the major cell type was replaced by memory T cells, mainly CD8[+] tissue resident memory T (CD8-TRM) cells and effector memory T (CD8-TEM) cells (Fig. 5d),

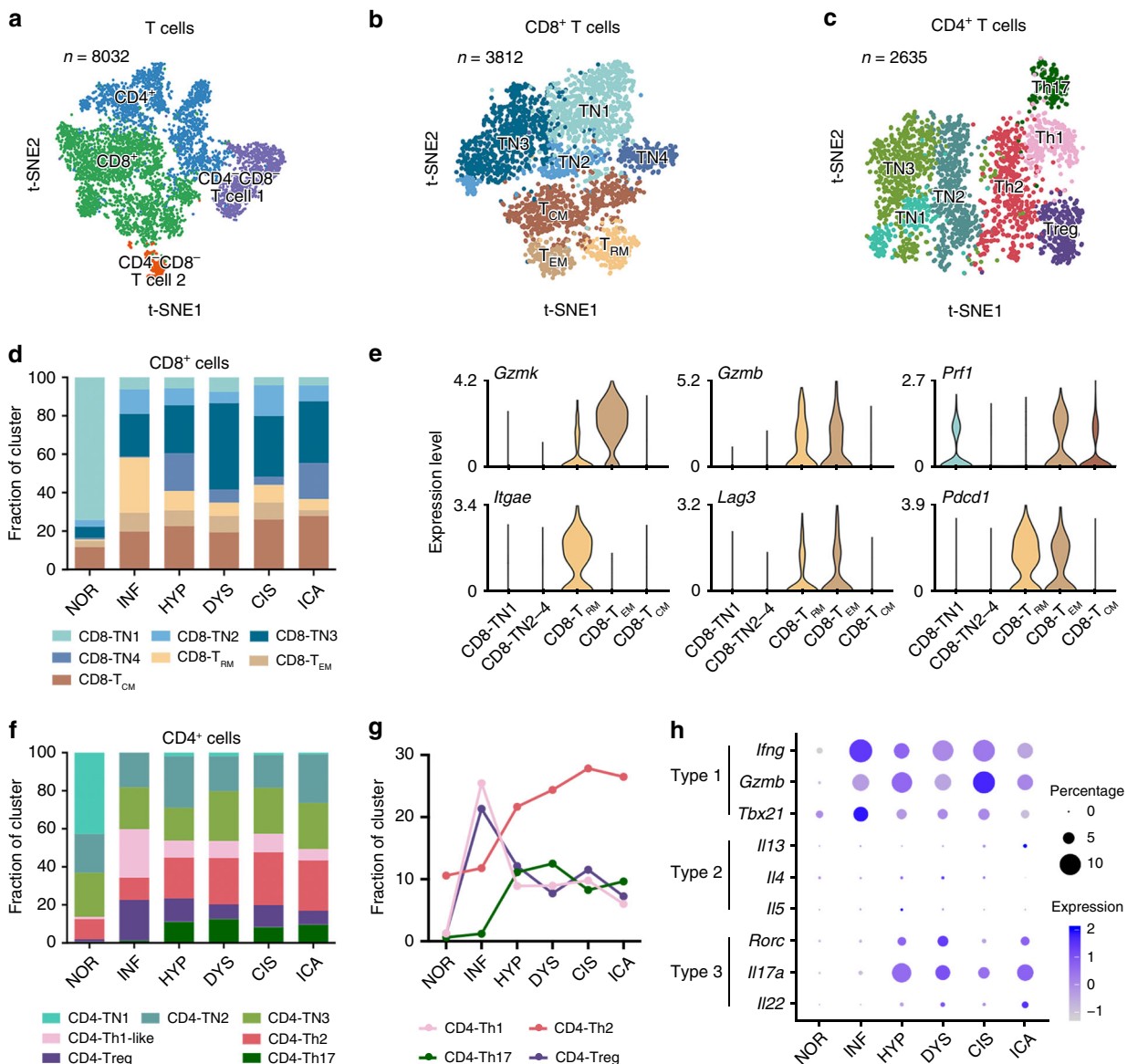

**Fig. 5 Characterization of multiple changes in T cell subtypes. a–c** tSNE plots of 8032 T cells, 3812 CD8$^+$ T cells, and 2635 CD4$^+$ T cells. Color indicates cluster. **d** Histogram showing CD8$^+$ T cell composition across the six pathological stages. Color indicates cluster designated in **b**. **e** Violin plots of marker gene expression among CD8$^+$ T cell clusters. **f** Histogram showing CD4$^+$ T cell composition across the six pathological stages. Color indicates cluster designated in **c**. **g** Line chart displaying changing trend of the four selected cluster proportions across the six pathological stages. **h** Bubble plot showing scale normalized expression of three representative genes involved in type 1, 2, and 3 immune response, respectively, from stage INF to stage ICA. Size of dots represents percentage of cells expressing corresponding genes in the stage.

suggesting that the immune responses were active in these two stages. Furthermore, we observed a high expression of the exhaustion markers (e.g., *Pdcd1* and *Lag3*)[42,43] in addition to the effective markers (e.g., *Gzmk, Gzmb,* and *Prf1*)[44,45] in these memory T cells (Fig. 5e); but the proportion of these memory T cells were declined along with tumorigenic processing after stage INF, indicating a non-effective CD8$^+$ T cell dominant microenvironment throughout precancerous stages.

Considering CD4$^+$ T cells (Fig. 5f; Supplementary Fig. 5b), we noticed an imbalance among CD4-Th1-like, CD4-Th2, and CD4-Th17 cells, involving in three major kinds of cell-mediated effector immunity, respectively[46]. Beginning at stage HYP, the proportion of CD4-Th1-like cells was decreased and replaced by CD4-Th2 and CD4-Th17 cells (Fig. 5f, g). CD4-Th1-like cells expressing high levels of *Tbx21*, *Gzmb*, and *Ifng* (Fig. 5h;

Supplementary Fig. 5c) had been shown to have anti-tumor capability through activating monocytes and responding to interferon-γ[46]. Differential gene expression analysis also revealed that some genes relative to immune activation (e.g., *Ifitm1, Ifitm2,* and *Rora*)[47,48] were significantly higher in CD4-Th1-like cells than in CD4-Th2 cells (Supplementary Fig. 5d), a type of immune cells that their role in tumorigenesis was uncertain. The activation of CD4-Th17 produced high levels of *Il22*, *Il17a*, and *Rorc* at stage HYP (Fig. 5h) that may activate monocytes and recruit neutrophils to respond to epithelial injuries[46]. The proportion of CD4-Th17 cells increased sharply after stage INF (Fig. 5f). The reduced anti-tumor role of CD4$^+$ T cells and increased inflammatory response suggest an immune response Type 1 to Type 3 transition[46] in the esophageal microenvironment during tumorigenesis.

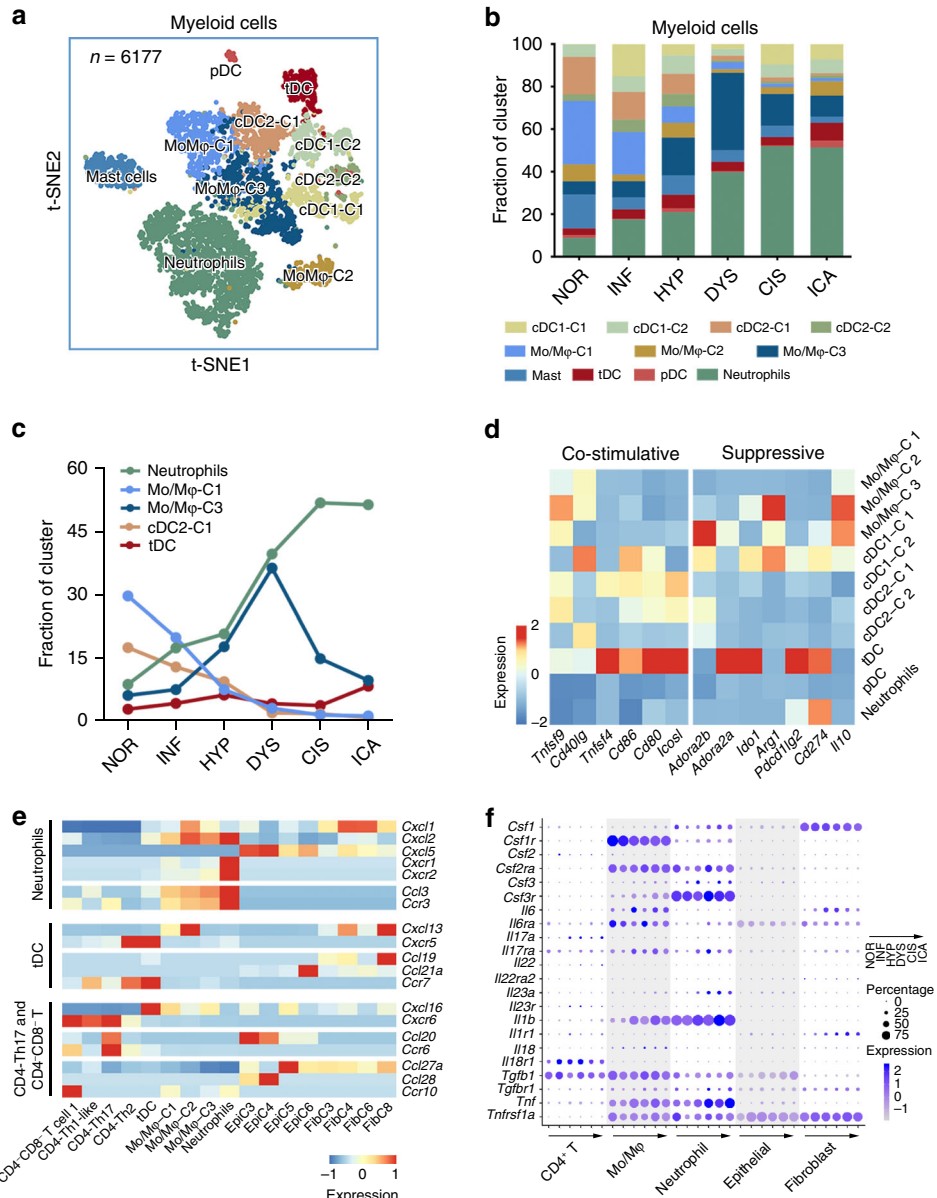

**Fig. 6 Compositional changes of myeloid cells and their interactions with other cells. a** tSNE plots of 6177 myeloid cells, colored by cluster. **b** Histogram showing myeloid cells composition across the six pathological stages. Color indicates cluster designated in **a**. **c** Line chart displaying changing trend of the five selected cluster proportions across the six pathological stages. **d** Heatmap showing scale normalized expression of the immune co-stimulation (left panel) or suppression (right panel) genes in the 11 clusters of myeloid cells. **e** Heatmap showing scale normalized expression of selected Ccl and Cxcl chemokines, and their corresponding receptors in representative clusters of epithelial cells, fibroblasts, myeloid cells, and T cells. **f** Bubble plot showing scale normalized expression of the selected genes along tumorigenesis process in various cell types. Size of dots represents percentage of cells expressing corresponding genes across pathological stages.

**Activated myeloid cells create inflammatory microenvironment.** We found that, among the 11 clusters of myeloid cells (Fig. 6a; Supplementary Fig. 6a), the proportions of monocyte/macrophage (Mo/Mφ)-C 1, Mo/Mφ-C 3, cDC2-C 1, tolerogenic dendritic cell (tDC), and neutrophils changed substantially along the stages of tumorigenesis (Fig. 6b, c). After stage INF, Mo/Mφ-C 1 cells were replaced by Mo/Mφ-C 3 cells, featured by decreased expression levels of the genes (e.g., H2-Ab1 and C1qa) relative to antigen presentation and complement (Supplementary Fig. 6b)[49,50]. Meanwhile, the expression level of genes involved in immunosuppression (e.g., Il10, Arg1, and Adora2b, Fig. 6d)[51–53] elevated along with the increase in Mo/Mφ-C 3 proportion. Furthermore, the proportion of cDC2-C 1 sharply decreased since stage NOR with the downregulation of Icosl, a gene representing

the immune stimulation activity[43] (Fig. 6d). In contrast, the immune suppressive tDC cells with high expressions of Cd274, Pdcd1lg2, and Ido1 were persisted throughout all the tumorigenic stages (Fig. 6d). In addition, we found that the number of neutrophils continuously increased, and these cells expressed high levels of Mmp9, Prok2, Vegfa, and Nos2, but not tumor suppressors (Supplementary Fig. 6c). These results suggested that myeloid cells were activated during the processes of ESCC tumorigenesis, resulting in an inflammatory microenvironment and CD8[+] T-associated adaptive immune suppression, which might allow initiated ESCC cells to survive and proliferate. Transcriptomic analysis also revealed that the immune suppression microenvironment might be created by the complex chemotaxis among various types of cells in this microenvironment.

For example, neutrophils recruitment could be mediated through its high expression of Cxcr1/2 receptors by their ligands Cxcl1/2/5 from other cell types or by their brother neutrophils through the simultaneous expression of Ccl3 and Ccr1. Similarly, tDC, Th17, and CD4−CD8− T cells could be recruited by chemokines secreted by different cell types (Fig. 6e).

Besides the roles of chemokines, cells in the esophageal microenvironment were also stimulated by each other through various signaling pathways. We found that from stage HYP, the expression levels of Il17a in CD4+ T cells and Il17ra in myeloid cells were elevated (Fig. 6f), which were well-known to have ability to activate macrophages and neutrophils[54]. On the other hand, we found significant correlations between Il1b and Il23a levels in macrophages and neutrophils and their receptors Il1r1 and Il23r in CD4+ T cells, implying close-loop activation among these cells. The activated immune cells secreted high levels of cytokines (Il1b, Il6, Il18, Tnf) and formed an inflammatory microenvironment starting from stage HYP (Fig. 6f). Analyses of interaction indicated increased interactions between malignant epithelial cells (EpiC 5 and EpiC 6) and inflammatory immune cells, Th17 and neutrophils (Supplementary Fig. 6d, e). The inflammatory microenvironment might further promote carcinogen-initiated epithelial cells to transit along the carcinogenic roadmap as illustrated in Figs. 2b and 4b.

**Validation of the expression profiles in human ESCC samples.** We last performed a validation examination of the identified expression profiles with mouse model in tissue samples obtained from four ESCC patients having both precancerous and cancerous lesions in the esophagus. RNA-seq was conducted with mini-bulk tissues of invasive carcinoma (ICA, n = 9), carcinoma in situ (CIS, n = 6), dysplasia (DYS, n = 17) and tumor-adjacent normal epithelia (taNOR, n = 8) obtained by LCM (Supplementary Fig. 7a). PCA showed a clearly different expression profile between cancer tissues (CIS and ICA) and non-cancer tissues (taNOR and DYS); however, an extensive overlap was present between taNOR and DYS (Fig. 7a; Supplementary Fig. 7b). In addition, we found that the six representative genes identified in mouse different epithelial cell clusters were also expressed in the precancerous lesions from ESCC patients (Fig. 7b) and the expression patterns were in line with those in mice. ALDH3A1 and S100A8 were highly expressed in the early precancerous lesions while MMP14 and ITGA6 were mainly present in invasive ESCC, which was concordant with their biological functions. As LCM captured mainly epithelial cells and the samples were not suitable for analyzing the status of other cell types, we performed bulk RNA-seq using biopsy samples. The results also showed a trend of increased neutrophils in the microenvironment, although this trend along the stage progress were marginally significant (trend test, P = 0.066; Fig. 7c). These results suggested that strongly supported our findings in mouse model and provided new clues in understanding ESCC initiation.

**Discussion**
In this study, we employed a high-throughput scRNA-seq analysis and have depicted a complete cell atlas of transition fates, characterized the transcriptomic features of cell clusters at each pathological stage, and elucidated the possible evolutionary trajectories of cells in the multistep processes of ESCC development. We majorly focused on the expression transitions of squamous epithelial and TME cells associated with tumorigenesis, because how they transform from normal to malignant phenotypes is a long-standing question. In this regard, our comprehensive findings shed light on the underlying molecular mechanism of carcinogen-induced esophageal squamous carcinoma.

Many previous studies have shown that human ESCC is a complex disease largely attributed to the exposure to carcinogens from the environment and lifestyles[55]. 4NQO represents a chemical carcinogen that can specifically induce ESCC in mice and this animal cancer model has been shown to mimic ESCC development in humans[6,7]. Thus, as it is impossible to obtain continuous tumorigenic lesions from the same human individuals over time, our results from this mouse model demonstrate its value for precisely understanding the gene expressions that may drive cell evolution from normal into malignancy in the multistep development of ESCC in humans. Indeed, we examine and validate the transcriptomic characters of epithelium and TME from patients bearing precancerous lesions.

Another interesting and important finding of this study is that the transcriptomic expression pattern of esophageal epithelial cells in 4NQO-induced early lesions is similar to that of esophageal normal epithelial cells in aged mice, while aging is a known risk factor for cancer development in humans. Previous genomic mutation studies in human esophagus[4] have shown that although the exonic mutation burden in normal esophageal epithelium derived from ESCC-free human individuals was increased with age, no cancer-related lesions occurred in view of histopathology, suggesting that the transcriptomic expression alterations and other factors such as immune suppression driven by carcinogen insults may show crucial effects in ESCC tumorigenesis.

Given that epithelial cells have two different fates, including oncogenic and differentiation, the deviation from the default fates during tumorigenesis implies a permissive or even supportive microenvironment for tumor development. Indeed, by analyze the dynamic changes of non-epithelial cell types in the microenvironment, we found decreased antigen presentation in both fibroblasts and myeloid cells reduced early downregulation of CD8+ T cell response during tumorigenesis. After inflammatory stage, no active cytotoxic responses by CD8+ T cells were detected indicating that neither epithelial cells nor fibroblasts in this pathogenic stage and hereafter had the activity tends to stimulate and activate cytotoxic T cells, whose activation has been considered to be a crucial mechanism against tumorigenesis. In addition, CD8+ T cell suppression may also result from a substantial change of myeloid cells because after pathogenic stage INF, MoMφ-C 3 and tDC cells, both of which are well-known as immune suppressors, became the main myeloid cell types in the tumor microenvironment. In addition, starting from hyperplasia stage, the macrophages and neutrophils activated by CD4-Th17 cells may form an inflammatory microenvironment during tumorigenesis. These findings strongly support our notion that the general immune suppression and chronic inflammatory microenvironment have crucial roles in promoting tumor development. Increased inflammatory responses and decreased cytotoxic T cell activities were also reported in the multistep process during cancer initiation for gastric cancer[56] and pancreatic cancer[57] depicted by single-cell analysis.

Although several important findings have been presented in this study, we acknowledge some limitations. First, the cells in this study were obtained from the mouse esophageal tissues induced by the carcinogen. Although the 4NQO-induced mouse ESCC model may mimic human ESCC development and we validated the expression of some genes and cells in human esophageal samples, caution should be taken in translating these results to humans. Second, although we have identified the carcinogen-induced epithelial cell and microenvironment transition fates, the underlying molecular mechanisms require further investigation. Furthermore, due to small size of esophageal epithelial tissues in mice and the vulnerable nature of esophagus epithelial cells in vitro, our data sets for early lesions were based

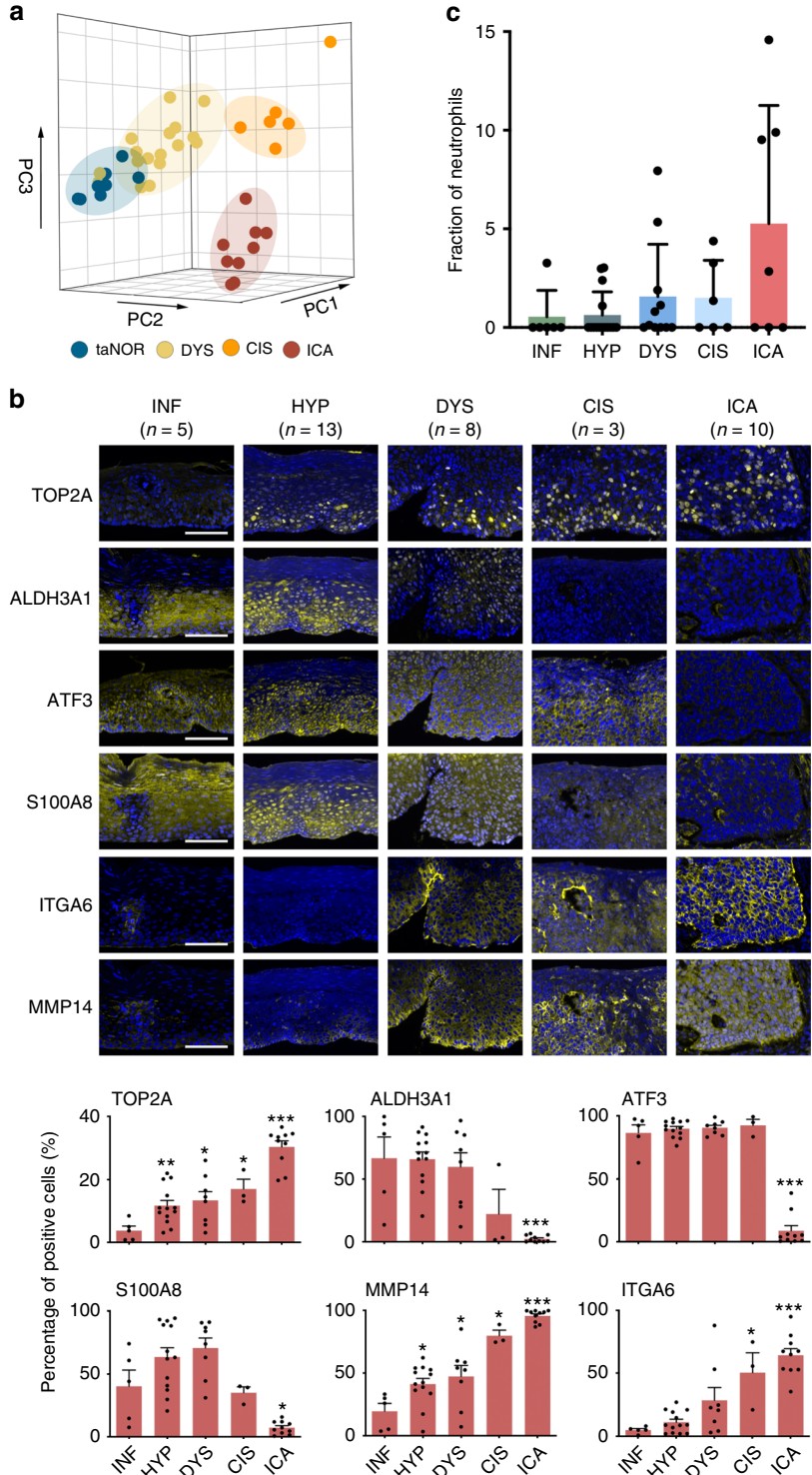

**Fig. 7 Validation of expression features in human epithelial tissues with different lesions. a** PCA plot of mini-bulk tissue RNA-seq on human esophageal tissues with different lesions as indicated by different colors and each dot represents a sample. **b** Immunofluorescence image (upper panel; scale bar, 100 μm) and quantification of selected gene expression levels (lower panel) for human epithelia tissue samples with different lesions. Independent samples from stage INF to ICA are $n = 5$, 13, 8, 3, and 10, respectively. Data represent mean ± S.E.M. *$P < 0.05$, **$P < 0.01$, ***$P < 0.001$, ****$P < 0.0001$ for two-sided Wilcoxon rank-sum test, compared with stage INF. $P$-values for TOP2A: 0.0098 (HYP), 0.0295 (DYS), 0.0357 (CIS), 0.0007 (ICA); ALDH3A1: 0.0007 (ICA); ATF3: 0.0007 (ICA); S100A8: 0.0127 (ICA); MMP14: 0.0140 (HYP), 0.0451 (DYS), 0.0357 (CIS), 0.0007 (ICA); ITGA6: 0.0357 (CIS), 0.0007 (ICA). **c** Proportion of neutrophils (relative to stage INF) in different precancerous and cancerous lesions based on bulk tissue RNA-seq using CIBERSORT. Each dot represents a sample, data represent mean ± S.D. $P = 0.066$, determined by two-sided Mantel–Haenszel chi-square test for linear trend. Independent patient samples from stage INF to ICA are $n = 6$, 15, 11, 6 and 7, respectively.

on relatively small amount of epithelial cells. This limitation could not have biased our conclusion because the major results were derived from the comparison between different clusters of epithelial cells rather than setting the precursor stage data set as a benchmark for the interpretation. However, experimental methods such as stripping the epithelial layer, reducing sample processing time and raising sensitivity of vulnerable cell detection should be improved in the future studies.

In conclusion, we have conducted the single-cell transcriptomic analysis of carcinogen-induced ESCC in mice, which mimics human ESCC development. By analyzing epithelial cells in different pathogenic stages, we have depicted a roadmap in carcinogen-induced mouse ESCC development and identified some key expression signatures. Comprehensive analysis of non-epithelial cells in tumor microenvironment revealed the suppression of anticancer immune response and robust expression of chronic inflammation genes. These results shed light on the transcriptomic alterations and transition status of various cell types in the esophagus when ESCC develops. Because the mouse model used in this study mimics human esophageal tumorigenesis, the findings may also represent human ESCC development.

## Methods

**Human biospecimen collection**. ESCC tumor, dysplasia lesions and tumor-adjacent (>5 cm) normal tissues of the same patients ($n = 4$) used for LCM and esophageal lesions of different pathological stages ($n = 45$) used for bulk RNA sequencing were collected during surgery or endoscopy in Linzhou Esophageal Cancer Hospital (Henan Province, China) from 2018 to 2019. The various lesions were diagnosed independently by at least two pathologists according to the American Joint Committee on Cancer Eighth edition. No patient had received chemotherapy or radiotherapy before biopsy or surgery. This study was approved by the Institutional Review Boards of Cancer Hospital, Chinese Academy of Medical Sciences and informed consent was obtained from each patient. Clinical information was collected from patients' medical records.

**Induction of multi-staged ESCC development and sample preparations**. Animal experiments in this study were conducted in compliance with approved protocols and guidelines from the Institutional Animal Care and Use Committee of the Chinese Academy of Medical Sciences. Eight-week-old female C57BL/6 mice, purchased from the Beijing Huafukang bioscience company in China, were maintained in local housing facility of a controlled condition (23 ± 1 °C, 50 ± 10% humidity and 12–12 h light-dark cycle). Mice were treated with 4NQO (Sigma-Aldrich) in drinking water (100 μg/ml) for 16 weeks to induce multi-staged ESCC carcinogenesis. Drinking water containing the carcinogen was replaced once a week with freshly prepared one and mice were allowed to access drinking water ad libitum during treatment. After 16 weeks of carcinogen treatment, 4NQO drinking water was replaced by sterile water until the mice were killed.

The esophageal lesions were identified by two independent pathologists based on the histopathological criteria described previously[58] (Fig. 1a). Briefly, stage NOR was well-oriented stratified epithelium consists of basal zone and superficial zone. Stage INF was normal epithelium with focal aggregates of epithelial lymphocytes. Stage DYS was defined as loss of polarity in the epithelial cells, nuclear pleomorphism, hyperchromatic, and increased or abnormal mitoses. In stage HYP, these abnormalities were confined to the lower third of the epithelium while in DYS they present in lower two thirds of the epithelium. Lesions with such abnormal changes involving the entire thickness of epithelium were considered as carcinoma in situ (stage CIS). Stage ICA was defined as a lesion with invasion into the sub-epithelial tissues. Esophageal samples of 4NQO-induced mice were subjected to single-cell RNA sequencing at six different time points and indicated different pathological stages: stage NOR (at week 0), stage INF (at week 12), stage HYP at (week 20), stage DYS (at week 22), stage CIS (at week 24), and stage ICA (at week 26). A group of control mice treated without 4NQO were killed at month 1, 2, 8, and 25 ($n = 2$, respectively). The esophagus was removed immediately when the animal was killed. Cross-sections of the esophagus were cut and stored at –80 °C and sections of the frozen tissues were stained with hematoxylin–eosin (H&E) for histopathological examination and microdissection for bulk RNA sequencing.

**Bulk RNA sequencing and data analysis**. Tissue samples from human or mice were cut into 5–10 consecutive sections (8 μm) and the epithelial layer contained 30–50 cells on each section was micro-dissected with a Leica LMD7000 laser-capture microdissection system. RNA was isolated from the mini-bulk samples and cDNA was prepared for sequencing based on the Geo-seq protocol[59]. Sequencing libraries were built using the TruePrep DNA Library Prep Kit V2 for Illumina (Vazyme), and evaluated by Bioanalyzer (DNA HS kit, Agilent). RNA-seq data were mapped to GRCh38 human genome and GRCm38 murine genome by

HISAT2 (version 2.1.0)[60] with default parameter for human and murine samples, respectively. The gene expression matrix of raw reads counts after annotation by HTSeq (version 0.6.1p1) was processed using the DESeq2 (version 1.22.2)[61] and visualized by showing the first 3 dimensions calculated by *plotPCA* function. We used TRIZOL to extract bulk RNA from patient biopsy ($n = 45$), and constructed sequencing library using NEBNext Ultra II RNA Library Prep Kit for Illumina. Sequencing data were processed by HISAT2 and HTseq as described above. The normalized expression from bulk samples of human precancerous lesions was used to estimated neutrophil fraction with CIBERSORT (version 1.06)[62].

**Single-cell RNA sequencing (scRNA-seq) and data analysis**. For mice at stage NOR and INF, the whole esophagi were taken and for mice at other stages, the dysplastic/malignant lesions were taken immediately after killed. The tissue samples of ESCC and various precursor lesions were gently minced into small pieces and digested for with in RPMI-1640 medium (Invitrogen) containing collagenase IV (Gibco) and hyaluronidase (Sigma-Aldrich). CD45-FITC (553080, BD Biosciences, dilution 1:20) antibody staining was performed for fluorescence activated cell sorting (FACS) on a FACSAria sorter (BD Biosciences). Single cells with or without GFP signal, representing immune or non-immune cells, were sorted and captured respectively in nanoliter droplets using Chromium (10× Genomics). scRNA-seq libraries were prepared using Chromium Single Cell 5′ Reagent Kits (10× Genomics) and sequencing was accomplished with an Illumina HiSeq x10 System.

Raw gene expression matrices obtained per sample using CellRanger (version 2.1.0, 10× genomics) were combined using the Seurat R package (version 2.3.4)[63]. Genes detected in <0.1% of all cells were filtered. We further excluded cells with gene counts <500 and cells that had >10% of mitochondrial gene expressions. After quality control, 66,089 cells were further analyzed for their gene expression profiles. The genes with normalized expression between 0.0125 and 3, and dispersion >0.5 were selected as highly variable genes. The resultants were first summarized by principle component analysis (PCA) and then first several PCs were selected for tSNE dimensional reduction using the default settings of the RunTSNE function. The numbers of resulting highly variable genes and the select PCs are shown in Supplementary Table 2. Cell clusters in the resulting two-dimensional representation were annotated as known biological cell types using canonical marker genes.

**Major cell-type clustering and marker gene identification**. We reanalyzed epithelial cells and stromal cells separately to identify their sub-clusters by using highly variable gene identification and dimensional reduction as described above (Supplementary Table 2). Cells with mix features were removed from further analysis (e.g., cells with both *Cd3d* and *Cd19* expression indicating T cell and B cell Multiplet). Clusters were identified using *FindClusters* function, and the specific gene markers for each cluster were determined using the *FindAllMarkers* function implanted in Seurat package.

**Gene set variation analysis (GSVA)**. Pathway analyses were predominantly performed on the 50 hallmark pathways described in the molecular signature database, exported using the MSigDB database (version 6.2)[64]. We also assessed biological process activities using a described biological process of Gene Ontology (GO) dataset. To assign pathway activity estimates to individual cells, we applied the GSVA using standard settings, as implemented in the GSVA package (version 1.30.0)[65]. To asses differential activities of pathways between sub-cluster of cells, we contrasted the activity scores for each cell using Limma package (version 3.38.3)[66]. Differential activities of pathways were calculated for each identified cluster. *T*-values of the results of some significant differential pathways ($P < 0.05$) in top 10 were visualized using heatmaps with average pathway activity scores of each cluster.

**Analysis of transcription factor expression**. SCENIC (version 1.1.0)[67] was used to assess the transcriptional activity of epithelial cells with high quality (UMI > 5500). The analysis used the motifs database for RcisTarget and GRNboost (corresponding to GENIE3 1.4.3, AUCell 1.4.1 and RcisTarget 1.2.1; with mm10__refseq-r80__10kb_up_and_down_tss.mc9nr). The input matrix was read counts.

**Cell transition trajectory and diffusion map analysis**. Monocle 2 (version 2.10.1)[68] was used for the trajectory analysis on high quality epithelial cells (UMI > 5500). All the top 100 markers of each cluster were used for the cell ordering. Dimensionality reduction and trajectory construction were performed on the selected genes with default methods and parameters. We calculated diffusion components using the *RunDiffusion* function as implemented in Seurat package with default parameters. The first three dimensions were used to draw diffusion maps. Mean coordinates of all of each cluster's cells were considered as the center of the cluster. The farthest cell of stage NOR from the total distance of other stages was the start point.

**Analysis of interaction between cell types**. We used CellPhoneDB (version 2.0.6)[69] with default arguments to reveal interaction between cell types. Each cluster analyzed was downsampled to 100 cells since the low cell number of some epithelial cluster. Interactions of epithelial cells with immune cells and immune cells with epithelial cells were demonstrated respectively.

**Immunohistochemistry and immunofluorescent detection**. Formalin-fixed paraffin-embedded (FFPE) sections of esophageal precursor lesions were collected from 30 patients between 2016 and 2018 in Linzhou Cancer Hospital, including INF ($n = 5$), HYP ($n = 13$), DYS ($n = 8$) and CIS ($n = 3$), to validate the results obtained in mice. The protein expression levels of the marker genes were detected by IHC staining for mice tissues and immunofluorescence for human specimens with antibodies (Abcam) shown in Supplementary Table 3. The samples were incubated with antibody against Ki67 (1:50 for IHC, ab16667), Top2a (1:8000 for IHC, 1:10,000 for IF, ab52934), Aldh3a1 (1:200 for IHC, 1:600 for IF, ab76976), Atf3 (1:200 for IHC, 1:600 for IF, ab216569), S100a8 (1:500 for IHC, 1:1500 for IF, ab92331), Mmp14 (1:2000 for IHC, 1:6000 for IF, ab51074), or Itga6 (1:250 for IHC, 1:750 for IF, ab181551). Opal multiplex staining was performed according to the Opal 5-Color Manual IHC Kit (Perkin Elmer). Opal DAPI, Opal 520, Opal 570, Opal 620, and Opal 690 were used to generate different signals. Slides were counterstained with DAPI (1:2000) for nuclei visualization, and subsequently coverslipped using a VectaShield Hardset mounting media. The slides were imaged using Vectra Polaris Automated Quantitative Pathology Imaging System (Perkin Elmer). We used inForm software (Perkin Elmer) to unmix and remove auto-fluorescence and to analyze the multispectral images.

**Statistical analysis**. Statistical analyses were conducted by using R v3.5.1[70] and Prism 7 (Graphpad Software). Pearson's correlation was calculated with the R function cor() and the significance was determined using two-sided unpaired Wilcoxon rank-sum test. $P < 0.05$ was considered statistically significant.

**Reporting summary**. Further information on research design is available in the Nature Research Reporting Summary linked to this article.

## Data availability

The raw sequencing data and processed gene expression matrix of mouse model have been deposited in GSA (Genome Sequence Archive in BIG Data Center, Beijing Institute of Genomics, Chinese Academy of Sciences, http://gsa.big.ac.cn) under the accession number CRA002118. The raw sequencing data of human esophageal tissues has been deposited in GSA-Human (https://bigd.big.ac.cn/gsa-human) under the accession number HRA000093. All the other data supporting the findings of this study are available within the article and its supplementary information files and from the corresponding author upon reasonable request. A reporting summary for this article is available as a Supplementary Information file. Source data are provided with this paper.

## Code availability

Example scripts to process and analyze data are available at https://github.com/ESCCemAll/scESCC_mice. Detailed information will be available from the corresponding authors upon reasonable request.

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

## Acknowledgements

This project was supported by the National Natural Science Foundation of China (81725015 to C. Wu, 21675098 to J.W., 21927802 to J.W.), Medical and Health Technology Innovation Project of Chinese Academy of Medical Sciences (2016-I2M-3-019 to D.L., 2016-I2M-4-002 to C. Wu, 2019-I2M-2-001 to D.L. and C. Wu), Beijing Outstanding Young Scientist Program (BJJWZYJH01201910023027 to C. Wu), 2018 Beijing Brain Initiative (Z181100001518004 to J.W.), Beijing Advanced Innovation Center for Genomics, and Beijing Advanced Innovation Center for Structural Biology.

## Author contributions

J.W., C. Wu., and D.L. conceptualized and supervised this study. J.Y., Q.C., W.F., Y.C., and T.L. contributed to the study design and performed most experiments. W.G., Y.X., and L.L. were engaged in sample collection and preparation. Y.M., J.Y., X.Z., and Y. Lin contributed to bioinformatics and statistical analysis. C. Wang, L.P., Y. Luo, and A.L. were responsible for animal experiments. J.Y., Q.C., W.F., Y.M., and Y.C. drafted the manuscript. W.T., C. Wu, J.W., and D.L. reviewed and prepared the final manuscript. All authors approved the final manuscript.

## Competing interests

The authors declare no competing interests.
