## [Peer Review File · Nature Communications]

Reviewers' comments:

Reviewer #1 (Remarks to the Author): Expert in GI cancer

The paper by Yao and colleagues examines 10X single cell RNAseq data from a mouse model for esophageal squamous carcinogenesis induced through the well known carcinogen 4NQO. Esophageal squamous cell cancer is a cancer that develops through chronic carcinogen exposure so it is fitting that Yao and colleagues opt to examine this model system since, potentially, it should mimic many of the features of the human condition. This carcinogen has been used in various contexts making it a reliable model system.

The manuscript is generally well written and easy to follow. A few minor spelling quirks (for example defferent line 703, animal mod line 79) should be easy to amend. Line 96 states that a mice of 25 months can be compared to an 80 yr old individual and this should be removed because the differences between individuals are simply too vast to make such sweeping statements. The figures and legends are well laid out and show the relevant statistical tests where appropriate.

I will evaluate the histopathology in the paper because I feel this is most within my area of expertise.

1) The authors do not include any data on tumour burden over time in these animals, meaning that readers cannot evaluate how efficient the carcinogen treatment protocol really is in their hands. This should be included.

2) At several crucial junctures the authors state they analysed scRNAseq 'on various esophageal lesions in 4NQO-exposed mice' (e.g. line 100). At no point do the authors make clear how many lesions were analysed for each stage. This should be included.

3) The various pathological stages were examined on frozen section. Comparing these histopathological stages on frozen section is notoriously difficult. The authors should include a sensitivity analysis to indicate what the impact of excluding or lumping subjective precursor stages such as dysplasia and CIS would mean to the results.

4) Can the authors explain why their data comprise only a minority of epithelial cells across the stages in comparison to fibroblasts and CD45+ populations. This is especially true for normal epithelium (Fig 1C) with hardly any normal epithelial cells included. It would appear this creates a significant issue in setting a benchmark for the interpretation of the precursor stage datasets.

Marnix Jansen, UCL

Reviewer #2 (Remarks to the Author): Expert in single cell transcriptomics

In this work, Yao et al conducted a single cell transcriptome study on diverse cell types along multi-step pathogenic process during ESCC initiation and development in a mouse model induced by 4NQO. The authors have demonstrated several interesting findings, and data are indeed quite informative and useful for understanding dynamic changes during ESCC carcinogenesis at the molecular level. Nevertheless, there are several major concerns, in particular as to the proportion biases of cell types and how to integrate the current findings with previously established findings. Also immunostained images are not quite high quality and should be improved. Detailed points are listed below.

Major concerns:

1) We noted that almost 88% CD45-nonimmune cells were fibroblasts in Figure 1c and 4a, far higher than epithelial cells. How to explain it? Is it due to experimental bias? More epithelial cells are required to reduce the impact of cell proportion biases.

2) We found that there existed apparent inconsistency between results shown in figure 2e and figure 2f. For example, TOP2A was found to be significantly upregulated in the INF phase within the single-cell atlas, while not showing any significant differences across phases in figure 2f. Similarly, inconsistencies also occurred for markers such as MMP14 and ATF3. In addition, evaluation of the consistency of these staining results within different samples in the same phase should be provided here.

3) Based on Fig2a & b, EpiC1 should represent proliferative cells. It was further found the normal phase harbored higher proportion of proliferative cells, and their proportion gradually decreased during ESCC carcinogenesis. This observation seemed to not be agreement with previous studies emphasizing proliferation as pivotal hallmarks underlying carcinogenesis. How to explain it?

4) In Figure3, pseudo-time trajectory analysis revealed two differentiation cell fates during the epithelial cell status transition, and some cells would evolved into cancer-like cells along the Component1. It is an interesting finding. However, from the pseudo-time trajectory we found the start point of the Component1 seemed to be the differentiated cell population EpiC4? It means that some EpiC4 cells would transdifferentiate into stem-like cells with potential as evolving to ESCC? More clear and detailed explanation should be provided herein.

5) Also in Figure 3, a transitional signature was identified to be associated with oncogenic evolution of the epithelial cells. Does these signatures included regulatory factors such as TFs, in particular as those TFs previously implicated to be associated with ESCC carcinogenesis? Moreover, is there any relationship between this signature and well-documented ESCC-related mutation or methylation dysregulation that indicated in the Introduction section? A detailed analysis needs to be provided.

6) As one of the authors' major findings, they found that the immune response type 1 was transited into type 3 during ESCC carcinogenesis. What is the biological basis for Type 1, Type 2, and Type 3 classifications, respectively?

7) Could there exist cross-talks between the inflammatory microenvironment and epithelial cells mentioned in this study? Which epithelial cell subtypes could be affected by the microenvironment? How to validate it?

8) Previous studies have depicted the single-cell atlas underlying the multi-step process during cancer initiation for other digestive tract organs, such as stomach (PMID: 31067475, GSE134520) and pancreas (PMID: 30385653). Comparison analysis is recommended.

Minor concerns:

1) We found that there existed great differences of the number of variable genes and UMIs between immune cells and epithelial cells. Why?

2) What does the horizontal coordinates of Fig 3b mean? The two upper and lower panels in Fig 3c are confused.

3) The coordinates in Figure 4d does not show completely.

Reviewer #3 (Remarks to the Author): Expert in immune landscapes

Yao et al. has elegantly demonstrated that the early-stage tumorigenesis can be studied through single cell RNA sequencing, using a mouse-model with esophageal damage. Unlike most single-cell studies of cancers, which capture only the mid- and late-stages of tumor development, this mouse ESCC model allows for uniquely capturing the very early stages of damage-induced inflammation and the onset of tumorigenesis in lesions. This work unveils many insights from a large-scale single cell

RNA-seq data, especially the relationship between cancer genesis and epithelium damage, through cell typing and identification of key expression signatures of epithelial cells. They specifically discussed the signatures related to inflammation, and linked to the further investigation of the inflammatory responses in tumor microenvironment by analyzing the fibroblasts and immune cells. Acquiring such a full picture of the microenvironment, and the interplay between various types of cells, can help better understanding the transition between inflammation and cancer, and the initiation of tumorigenesis. This is an important work. However, I have several questions and would like to ask authors to address and clarify them.

Major points:

1. The authors should elaborate more on the rationale of using such a damage-model of ESCC over other models based on transgenic animals. This damage model is a good model for oncogenesis of many cancers such as ESCC, however, I haven't seen many papers using it. I suggest the authors to discuss on the justification of this model.
2. Figure 2d is interesting. Through this figure, one can probably see the progress of damage-induced inflammatory responses, and the trend towards tumorigenesis. However, I feel this figure is not easy to follow, probably due to the 2D projection creates too much overlap on the trajectories. Is it possible to add one more dimension, or use other types of illustration, to better depict this result?
3. It is also very important for authors to provide sufficient histology data, such as the IHC images, of animal models and of human with different ages. Without such experimental evidence, false analyses may be drawn from the single cell sequencing data and lead to wrong conclusion. I would suggest the authors to re-arrange the IHC data, maybe putting both animal model and human data together, and to discuss them together, to strengthen the effectiveness of the animal model.
4. It is interesting to see the number of epithelial cells are rare in the control samples. This observations seems similar to many other reports. Why is that? Is it because epithelial cells are so rare in normal tissues? Can authors provide an explanation?

Minor points:

1. This manuscript contains a large-volume of data with many figures/panels. Maybe some panels, or even figures, can be rearranged and/or combined to be more concise?
2. Fig 7a and 7c are from two sets of data, but the discussion is limited in the main text. I also think using two sets of data may be a little bit redundant. Probably it is better to elaborate both data sets with some details. In addition, Fig 7b should take up more space in the plot compared to other panels.

Responses to the Reviewers' Comments

Reviewer #1

General comment

The paper by Yao and colleagues examines 10X single cell RNAseq data from a mouse model for esophageal squamous carcinogenesis induced through the well known carcinogen 4NQO. Esophageal squamous cell cancer is a cancer that develops through chronic carcinogen exposure so it is fitting that Yao and colleagues opt to examine this model system since, potentially, it should mimic many of the features of the human condition. This carcinogen has been used in various contexts making it a reliable model system. The manuscript is generally well written and easy to follow. A few minor spelling quirks (for example defferent line 703, animal mod line 79) should be easy to amend. Line 96 states that a mice of 25 months can be compared to an 80 yr old individual and this should be removed because the differences between individuals are simply too vast to make such sweeping statements. The figures and legends are well laid out and show the relevant statistical tests where appropriate.

Response: Thank you for these positive comments. We have edited out all spelling typos throughout the manuscript text. Per the comment, we have removed "equivalent to a man aged 80 years" in revised manuscript (page 5, line 110).

I will evaluate the histopathology in the paper because I feel this is most within my area of expertise.

Specific Comments

Comment 1. The authors do not include any data on tumour burden over time in these animals, meaning that readers cannot evaluate how efficient the carcinogen treatment protocol really is in their hands. This should be included.

Response 1: We appreciate the suggestion. All mice receiving 4NQO developed the expected lesions including ESCC in the esophagus. Tumor number per animal at the end of experiment (Week 26) was 6.0 ± 3.6 (mean \pm SD). These results demonstrate that our carcinogen treatment protocol is efficient in inducing ESCC and its precursor lesions in these mice. Per the suggestion, we have provided the data on tumor burden in 4NQO treated mice (page 5, lines 99-102). By the way, we have successfully established this carcinogen induced mice ESCC model in our previous published study, which has been cited in revised manuscript (ref 19: Chu et al, Theranostics 2020). The carcinogen treatment protocol in the present study was the same as previously published.

Comment 2. At several crucial junctures the authors state they analysed scRNAseq 'on various esophageal lesions in 4NQO-exposed mice' (e.g. line 100). At no point do the authors make clear how many lesions were analysed for each stage. This should be included.

Response 2: We used whole mouse esophagus that were normal (NOR, n=30) or at Stages INF (n=20). For other lesions, the sample numbers were 32 HYP from 25 mice, 24 DYS from 17 mice, 24 CIS from 20 mice and 25 ICA from 23 mice. A brief statement has also been added to the text (page 5, lines 114-117).

Comment 3. The various pathological stages were examined on frozen section. Comparing these histopathological stages on frozen section is notoriously difficult. The authors should include a sensitivity analysis to indicate what the impact of excluding or lumping subjective precursor stages such as dysplasia and CIS would mean to the results.

Response 3: We agree that comparing the histopathological stage of precursor lesions such as hyperplasia and dysplasia is difficult on frozen section. However, we did observe the pathological progression over time in this mouse model as shown in Figure 1a by repeating the mouse model experiments and examining the pathological change by two pathologists independently. Per the suggestion, we have performed a pseudotime analysis by excluding one stage at a time and the results are similar to that without excluding the stages such as hyperplasia and dysplasia (Supplementary Fig.3i, j). We believe these results mean to our conclusion. Accordingly, we have added two sentences to Results section (pages 9, lines 206-210). We have also added the statements of lesion identification to Methods section (page 19, lines 430-439).

Comment 4. Can the authors explain why their data comprise only a minority of epithelial cells across the stages in comparison to fibroblasts and CD45+ populations. This is especially true for normal epithelium (Fig 1c) with hardly any normal epithelial cells included. It would appear this creates a significant issue in setting a benchmark for the interpretation of the precursor stage datasets.

Response 4: We detected 1,756 epithelial cells in all samples. In Stage NOR samples only 20 epithelial cells were obtained. This result might be attributable to: (1) the mouse esophageal epithelium is very small in size (3-4 cm long and 40-50 μ m thick), which can provide only limited epithelial cells for analysis; (2) normal epithelial cells are more vulnerable than other cell types such as fibroblasts in vitro (Chopra et al. *Stem Cells Dev* 2010;19:131–142; Schlaermann et al. *Gut* 2016;65:202–213); (3) physical damage during single-cell suspension preparation and FACS might also reduce the viability of normal epithelial cells. Together, these reasons might have finally caused epithelial cells as the minority across the stages in comparison to fibroblasts and CD45+ populations. We therefore pooled the esophagi at Stage NOR from 30 mice to obtain enough epithelial cells. We did not use the normal epithelial cells as a benchmark for the interpretation of the precursor stage datasets. Instead, most conclusions were derived from the comparison between different clusters of epithelial cells.

We have briefly discussed this issue as a limitation (page 17, lines 391-396), and hope these revisions are satisfactory.

Reviewer #2

General comment

In this work, Yao et al conducted a single cell transcriptome study on diverse cell types along multi-step pathogenic process during ESCC initiation and development in a mouse model induced by 4NQO. The authors have demonstrated several interesting findings, and data are indeed quite informative and useful for understanding dynamic changes during ESCC carcinogenesis at the molecular level. Nevertheless, there are several major concerns, in particular as to the proportion biases of cell types and how to integrate the current findings with previously established findings. Also immunostained images are not quite high quality and should be improved. Detailed points are listed below.

Response: Thanks for the comments. Per the suggestion, the immuno-stained images have been replaced with high quality ones (Fig 2f and Fig 7b). We address other concerns as follows.

Major concerns:

Comment 1: We noted that almost 88% CD45-nonimmune cells were fibroblasts in Figure 1c and 4a, far higher than epithelial cells. How to explain it? Is it due to experimental bias? More epithelial cells are required to reduce the impact of cell proportion biases.

Response 1: A similar comment is also raised by Reviewer #1 and we have addressed it. It is true that the proportion of epithelial cells is relatively small, especially in samples with Stage NOR. This result might be attributable to: (1) the mouse esophageal epithelium is very small in size (3-4 cm long and 40-50 μm thick), which can provide only limited epithelial cells for analysis; (2) normal epithelial cells are more vulnerable than other cell types such as fibroblasts in vitro (Chopra et al. 2010; Schlaermann et al. 2016); (3) physical damage during single-cell suspension preparation and FACS might also reduce the viability of normal epithelial cells. Together, these might have finally caused epithelial cells as the minority across the stages in comparison to fibroblasts and CD45+ populations. We have therefore pooled the esophagi with Stage NOR from 30 mice to obtain enough epithelial cells. We did not use the normal epithelial cells as a benchmark for the interpretation of the precursor stage datasets. Instead, most conclusions were derived from the comparison between different clusters of epithelial cells.

We have briefly discussed this issue as a limitation (page 17, lines 391-396), and hope these revisions are satisfactory.

Comment 2: We found that there existed apparent inconsistency between results shown in figure 2e and figure 2f. For example, TOP2A was found to be significantly upregulated in the INF phase within the single-cell atlas, while not showing any significant differences across phases in figure 2f. Similarly, inconsistencies also occurred for markers such as MMP14 and ATF3. In addition, evaluation of the consistency of these staining results within different samples in the same phase should be provided here.

Response 2: We appreciate the comment. Per the suggestion, we have changed Fig. 2e to show average level of each gene expression across stages, which is easy to compare with their protein levels in Fig. 2f. Top2a was highly expressed in EpiC 1 and represents proliferating epithelial cells. We observed that Top2a was stably expressed over time at both RNA and protein levels. Atf3 RNA expression level peaked at Stage INF and then decreased gradually along the advancing stages. A similar trend was observed for Atf3 protein level detected by IHC. Mmp14 RNA was expressed in all stages and the level was slightly increased at Stage ICA. IHC analysis also showed the highest Mmp14 protein level at Stage CIA, although the mRNA and protein levels are not that consistent along other stages. It is well known that for some genes in some organs and tissues the mRNA level is not always consistent with the protein levels, which could be due to possible post-transcriptional modification and other biological processes. It could be the case at Stage NOR esophageal epithelium, where Mmp14 mRNAs are produced but the protein may not need. However, for carcinogenic epithelial cells the protein is translated because of need of these cells. To more precisely and clearly describe these results, we have reworded the statements in the Results section of revised manuscript (page 7-8, lines 163-172). Per the suggestion, we have provided all images of IHC on different samples in Supplementary Fig. 2d.

Comment 3: Based on Fig2a & b, EpiC1 should represent proliferative cells. It was further found the normal phase harbored higher proportion of proliferative cells, and their proportion gradually decreased during ESCC carcinogenesis. This observation seemed to not be agreement with previous studies emphasizing proliferation as pivotal hallmarks underlying carcinogenesis. How to explain it?

Response 3: Among the 6 clusters of epithelial cells, EpiC1 represents a common component in all pathological stages. Pathway analysis of gene expression in EpiC 1 showed that EpiC 1 are cells that have self-renewal function and are responsible for homeostatic proliferation in normal esophageal epithelium, suggesting that EpiC 1 cells are normal cells. It is reasonable and expected

that the proportion of EpiC 1 cells gradually decreased, while the proportions of malignant cells such as EpiC 5 and EpiC 6 gradually increased during ESCC carcinogenesis. It is the proliferation of malignant cells (e.g., EpiC 5 and 6) that is the pivotal hallmark of carcinogenesis.

Comment 4: In Figure 3, pseudo-time trajectory analysis revealed two differentiation cell fates during the epithelial cell status transition, and some cells would evolved into cancer-like cells along the Component 1. It is an interesting finding. However, from the pseudo-time trajectory we found the start point of the Component 1 seemed to be the differentiated cell population EpiC 4? It means that some EpiC 4 cells would transdifferentiate into stem-like cells with potential as evolving to ESCC? More clear and detailed explanation should be provided herein.

Response 4: Based on pseudo-time trajectory analysis, EpiC 1 had the lowest pseudo-time value among all the clusters, suggesting that they are the start point of cell state transition. EpiC 4 cells, which had higher pseudo-time value, are not the start point of state transition. Instead, EpiC 4 cells could be the normally differentiated epithelial cells. We identified two fates of esophageal epithelial cells during ESCC tumorigenesis, both starting from EpiC 1: some EpiC 1 cells, which might not be attacked by the carcinogen, finally became normally differentiated EpiC 4, but other EpiC 1, which were initiated by the carcinogen, finally became malignant EpiC 6 cells, processing through EpiC 2 to EpiC 5 cells. We have reworded the text in Results section (page 8, lines 174-178) and rearranged Fig. 3a to make the relevant statements and resultant figure clearer (Supplementary Fig. 3a). We hope this revision is satisfactory.

Comment 5: Also in Figure 3, a transitional signature was identified to be associated with oncogenic evolution of the epithelial cells. Does these signatures included regulatory factors such as TFs, in particular as those TFs previously implicated to be associated with ESCC carcinogenesis? Moreover, is there any relationship between this signature and well-documented ESCC-related mutation or methylation dysregulation that indicated in the Introduction section? A detailed analysis needs to be provided.

Response 5: Thanks for the thoughtful comment. We have analyzed the transcriptional activity along the oncogenic evolution of Component 1 in Fig. 3a using SCENIC (Diether L. et al, 2018) and identified several significant changes of TFs related to ESCC carcinogenesis and the results are presented in Supplementary Fig. 3e. We have found that the activities of TFs that are known to have tumor suppressor function, such as *Pitx1*, *Trp53* and *Bclaf1*, decreased along with Component 1, while TFs related to cell migration, invasion, EMT or angiogenesis, such as *Creb3* and *Elk3*, significantly increased along with Component 1. We have also analyzed the potential expression alterations of genes that previously reported methylated in ESCC and found that *Rab25* was down-regulated while *Met* was up-regulated along Component 1. Both genes are known to play important roles in promoting cancer cell migration and invasion. For well-documented ESCC-related mutation, we have found that the expression levels of tumor suppressor *Notch1* were gradually decreased over time along the evolution of Component 1.

Per the comment, we have added these results in revised manuscript (page 8-9, lines 185-195, Fig. 3d and Supplementary Fig. 3e). Accordingly, we have also added a small section "Analysis of transcription factor expression" to the Methods section (page 22, line 505-509).

Comment 6: As one of the authors' major findings, they found that the immune response type 1 was transited into type 3 during ESCC carcinogenesis. What is the biological basis for Type 1, Type 2, and Type 3 classifications, respectively?

Response 6: As you know, there are 3 major types of immunity: Type 1 consists of CD8+ cytotoxic T cells and CD4+ Th1 cells, characterized by producing interferon- γ and T-bet. Type 2 involves the participation of CD4+ Th2 cells, characterized by producing IL-4, IL-5 and IL-13 while Type 3 is mediated by Rorc+ Th17 cells producing IL-17 and IL-22 (Mosmann et al. 1986; Harrington et al. 2005; Annunziato et al. 2015). We have reworded the statement and cited these references which define the biological basis for Type 1, 2 and 3 classifications in revised manuscript (page 12, lines 274-276).

Comment 7: Could there exist cross-talks between the inflammatory microenvironment and epithelial cells mentioned in this study? Which epithelial cell subtypes could be affected by the microenvironment? How to validate it?

Response 7: Per the comment, we have analyzed the cross-talk between inflammatory microenvironment and epithelial cells and identified concordant expressions of several chemokine ligand and receptor pairs, which suggest the interactions. We found that malignant epithelial cells produced Cxcl5 and Ccl20, which might recruit neutrophils and CD4+ Th17 cells, two cell types that play important roles in Type 3 immunity and inflammatory immune response at later stage of tumorigenesis. In addition, analysis of interaction of epithelial cells with microenvironment showed that EpiC 5 and EpiC 6 recruit and stimulate immune cells via CXCLs, CSF, TNF and their receptors; immune cells regulate EpiC 5 and EpiC 6 via Cd44, Icam1 and Ctla4. We have added these results and methods in revised manuscript (page 14, lines 311-313; Supplementary Fig. 6d, e; page 22, lines 519-523).

There exist several methods to validate the interactions, for instance, the immunofluorescence co-localization techniques. Unfortunately, we have difficult to perform validation analysis at this stage of research because it is unable to distinguish each cluster of epithelial cells on the section of esophageal tissues.

Comment 8: Previous studies have depicted the single-cell atlas underlying the multi-step process during cancer initiation for other digestive tract organs, such as stomach (PMID: 31067475, GSE134520) and pancreas (PMID: 30385653). Comparison analysis is recommended.

Response 8: We thank for the comment. Because of different organs (the esophagus, stomach and pancreas) and different etiology for tumorigenesis of these types of cancer, comparison of our findings with these studies might not really mean. However, per the comment, we have made a comparison analysis with those the Reviewer mentioned in the comment (Bernard et al. 2019 on pancreatic cancer and Zhang et al. 2019 on gastric cancer). Both studies reported an increased inflammatory response and a decreased cytotoxic T cell activity, which are comparable with our results in the present study. We have added a brief discussion and cited these two literatures in Discussion section (page 16-17, lines 381-383; Refs 86 and 87).

Minor concerns:

Comment 1: We found that there existed great differences of the number of variable genes and UMIs between immune cells and epithelial cells. Why?

Response 1: The reason is not immediately evident; however, the gene expressions were highly heterogeneous in epithelial cells during tumorigenesis than immune cells due to their direct exposure to the carcinogen and injuries, which leads to the higher number of variable genes in epithelial cells than in immune cells. We have added Supplementary Fig. 1b and 1c in revised manuscript, which shows that the number of genes and UMIs in non-immune cells is higher than

that in immune cells; the results are in line with previous study (Please refer Schaum et al. Nature 2018; 562:367-372; PMID: 30283141).

Comment 2: What does the horizontal coordinates of Fig 3b mean? The two upper and lower panels in Fib3c are confused.

Response 2: The horizontal coordinates of Fig. 3b is Component 1 value. We have rearranged Fig. 3b and 3c in revised manuscript. Thanks.

Comment 3: The coordinates in Figure 4d does not show completely.

Response 3: We have fixed it in revised manuscript. Thanks.

Reviewer #3

General comment

Yao et al. has elegantly demonstrated that the early-stage tumorigenesis can be studied through single cell RNA sequencing, using a mouse-model with esophageal damage. Unlike most single-cell studies of cancers, which capture only the mid- and late-stages of tumor development, this mouse ESCC model allows for uniquely capturing the very early stages of damage-induced inflammation and the onset of tumorigenesis in lesions. This work unveil many insights from a large-scale single cell RNA-seq data, especially the relationship between cancer genesis and epithelium damage, through cell typing and identification of key expression signatures of epithelial cells. They specifically discussed the signatures related to inflammation, and linked to the further investigation of the inflammatory responses in tumor microenvironment by analyzing the fibroblasts and immune cells. Acquiring such a full picture of the microenvironment, and the interplay between various types of cells, can help better understanding the transition between inflammation and cancer, and the initiation of tumorigenesis. This is an important work. However, I have several questions and would like to ask authors to address and clarify them.

Response: We appreciate the Reviewer for his/her elegant and very positive comments and are happy to address the following questions.

Major points:

Comment 1: The authors should elaborate more on the rationale of using such a damage-model of ESCC over other models based on transgenic animals. This damage model is a good model for oncogenesis of many cancers such as ESCC, however, I haven't seen many papers using it. I suggest the authors to discuss on the justification of this model.

Response 1: Numerous epidemiological studies have shown that ESCC is mainly caused by environmental chemical carcinogens. To elucidate the processes of ESCC carcinogenesis, the best and mimic way is to induce ESCC in animal models using a specific chemical carcinogen rather than using transgenic animals where the etiology is genetic changes such as gene mutations. 4NQO-induced ESCC mouse models have been well established in many labs including ours, which has been stated in Introduction section of our manuscript (page 4, lines 76-78).

Comment 2: Figure 2d is interesting. Through this figure, one can probably see the progress of damage-induced inflammatory responses, and the trend towards tumorigenesis. However, I feel this figure is not easy to follow, probably due to the 2D projection creates too much overlap on the trajectories. Is it possible to add one more dimension, or use other types of illustration, to better depict this result?

Response 2: Per the suggestion, we have added the 3D projection to Fig. 2d in revised manuscript Supplementary Fig. 2c and believe it is easy to follow now. Thanks.

Comment 3: It is also very important for authors to provide sufficient histology data, such as the IHC images, of animal models and of human with different ages. Without such experimental evidence, false analyses may be drawn from the single cell sequencing data and lead to wrong conclusion. I would suggest the authors to re-arrange the IHC data, maybe putting both animal model and human data together, and to discuss them together, to strengthen the effectiveness of the animal model.

Response 3: Per the comment, we have added IHC data in Supplementary Fig. 2d. As for the suggestion of re-arranging the IHC data, we would like to persist in not doing so because to do this will demolish our article structure.

Comment 4: It is interesting to see the number of epithelial cells are rare in the control samples. This observations seems similar to many other reports. Why is that? Is it because epithelial cells are so rare in normal tissues? Can authors provide an explanation?

Response 4: Please also refer our Response 4 to Reviewer #1 who raises a similar issue. We detected 1,756 epithelial cells in all samples. In Stage NOR samples only 20 epithelial cells were obtained. This result might be attributable to: (1) the mouse esophageal epithelium is very small in size (3-4 cm long and 40-50 μm thick), which can provide only limited epithelial cells for analysis; (2) normal epithelial cells are more vulnerable than other cell types such as fibroblasts in vitro (Chopra et al. 2010; Schlaermann et al. 2016); (3) physical damage during single-cell suspension preparation and FACS might also reduce the viability of normal epithelial cells. Together, these reasons might have finally caused epithelial cells as the minority across the stages in comparison to fibroblasts and CD45+ populations. We therefore pooled the esophagi at Stage NOR from 30 mice to obtain enough epithelial cells. We did not use the normal epithelial cells as a benchmark for the interpretation of the precursor stage datasets. Instead, most conclusions were derived from the comparison between different clusters of epithelial cells.

We have briefly discussed this issue as a limitation (page 17, lines 391-396), and hope these revisions are satisfactory.

Minor points:

Comment 1: This manuscript contains a large-volume of data with many figures/panels. Maybe some panels, or even figures, can be rearranged and/or combined to be more concise?

Response 1: Indeed this manuscript contains a lot of data including 7 text figures and 7 Supplementary figures. We have tried our best to arrange figures in the way for readers to understand this study. Some papers published in Nat Commun have 10 text figures.

Comment 2: Fig 7a and 7c are from two sets of data, but the discussion is limited in the main text. I also think using two sets of data may be a little bit redundant. Probably it is better to elaborate both data sets with some details. In addition, Fig 7b should take up more space in the plot compared to other panels.

Response 2: Yes. We provided two set of data: one from LCM samples which contain epithelium only and show the pattern across ESCC progress (Fig. 7a); another from bulk esophageal tissue samples which show detailed data of neutrophil fractions in different stages of ESCC

carcinogenesis (Fig. 7c). These data are necessary but not redundant. LCM samples capture mainly epithelial cells that are good for examining the status of epithelial cells but not suitable for analyzing the other cell types. In view of this fact, we have to perform RNA-seq using biopsy bulk samples (unfortunately, such biopsy samples are too small to do single-cell RNA-seq). Therefore, the two sets of data bring up to full strength. Per the suggestion, we have added a statement in revised manuscript (page 14, line 329-331). The size of Fig. 7b has been adjusted.

REVIEWERS' COMMENTS:

Reviewer#1:

The authors have satisfactorily addressed my concerns, with one impt exception: all three reviewers have indicated that the comparatively small number of epithelial cells analysed here is a concern. With this rebuttal the authors have not produced any novel data to reassure the reviewers on this point. Their argument that the small number of epithelial cells could not have influenced results is unconvincing since this remains a very small sample size meaning reproducibility is in question.

I dont think this is a fatal flaw. But the authors should clearly and plainly acknowledge this caveat to the results in the discussion of the manuscript and suggest improved methodology, ie this caveat should not be 'buried' in the discussion. Gently stripping the epithelial layer is easy and straightforward.

minor

1. misspelt 'caner' Fig 3a
2. 'similar to that reported previously'. It is unclear from this phrasing that these are the authors' 2020 data (ref 19), possibly on the same cohort of mice). This should be removed so as not to suggest external reproducibility.

Marnix Jansen, UCL London

Reviewer#2:

The authors have substantially improved their manuscript and addressed all my major comments. I think the current version is well written, with better organization and clearer explanation.

Reviewer#3:

I thank the authors for the extensive responses. All my concerns have been addressed.

RESPONSES TO THE REVIEWERS' COMMENTS

Reviewer #1:

The authors have satisfactorily addressed my concerns, with one impt exception: all three reviewers have indicated that the comparatively small number of epithelial cells analysed here is a concern. With this rebuttal the authors have not produced any novel data to reassure the reviewers on this point. Their argument that the small number of epithelial cells could not have influenced results is unconvincing since this remains a very small sample size meaning reproducibility is in question. I don't think this is a fatal flaw. But the authors should clearly and plainly acknowledge this caveat to the results in the discussion of the manuscript and suggest improved methodology, ie this caveat should not be 'buried' in the discussion. Gently stripping the epithelial layer is easy and straightforward.

Response: We appreciate the comment and, per the suggestion, have added a sentence 'However, experimental methods such as stripping the epithelial layer, reducing sample processing time and raising sensitivity of vulnerable cell detection should be improved in the future studies' to Discussion section in revised manuscript (lines 405–407). We hope this revision is satisfactory.

minor

1. misspelt 'caner' Fig 3a.

Response: We have fixed this typo in Fig. 3a.

2. 'similar to that reported previously'. It is unclear from this phrasing that these are the authors' 2020 data (ref 19), possibly on the same cohort of mice). This should be removed so as not to suggest external reproducibility.

Response: Thank you. Per the suggestion, we have removed this statement.

Reviewer #2:

The authors have substantially improved their manuscript and addressed all my major comments. I think the current version is well written, with better organization and clearer explanation.

Response: Thank you very much.

Reviewer #3:

I thank the authors for the extensive responses. All my concerns have been addressed.

Response: Thank you very much.